# Memory, switches, and an OR-port through bistability in chemically fueled crystals

Fabian Schnitter[1], Benedikt Rieß[1], Christian Jandl[2] & Job Boekhoven [1,3 ✉]

The ability to store information in chemical reaction networks is essential for the complex behavior we associate with life. In biology, cellular memory is regulated through transcriptional states that are bistable, i.e., a state that can either be on or off and can be flipped from one to another through a transient signal. Such memory circuits have been realized synthetically through the rewiring of genetic systems in vivo or through the rational design of reaction networks based on DNA and highly evolved enzymes in vitro. Completely bottom-up analogs based on small molecules are rare and hard to design and thus represent a challenge for systems chemistry. In this work, we show that bistability can be designed from a simple non-equilibrium reaction cycle that is coupled to crystallization. The crystals exert the necessary feedback on the reaction cycle required for the bistability resulting in an on-state with assemblies and an off-state without. Each state represents volatile memory that can be stored in continuously stirred tank reactors indefinitely even though molecules are turned over on a minute-timescale. We showcase the system's abilities by creating a matrix display that can store images and by creating an OR-gate by coupling several switches together.

[1] Department of Chemistry, Technical University of Munich, Lichtenbergstrasse 4, 85748 Garching, Germany. [2] Catalysis Research Centre, Technical University of Munich, Lichtenbergstrasse 4, 85748 Garching, Germany. [3] Institute for Advanced Study, Technical University of Munich, Lichtenbergstrasse 2a, 85748 Garching, Germany. ✉email: job.boekhoven@tum.de

An outstanding goal for the field of systems chemistry is the ability to record information of events that occurred in the past in chemical reaction networks, i.e., networks that can store memory[1–5]. In biology, such volatile memory is stored in transcriptional states that are bistable, i.e., a gene is either expressed or not and can be switched from one to another through a transient signal[6,7]. Because most molecules in the cell are constantly degraded and resynthesized (i.e., turned over), this form of memory is dynamic, i.e., the molecules involved are continuously sustained in a steady state[6]. Synthetic counterparts of cellular memory have been realized through the rewiring of genetic systems in vivo or through the rational design of reaction networks based on DNA and highly evolved enzymes in vitro[8–13]. These networks rely on autocatalytic chemical reaction systems, i.e., reactions in which the product accelerates its own production. These autocatalytic reactions in biological systems rely on highly evolved enzymes. Non-enzymatic reaction networks that display bistability also exist but rely on autocatalytic reactions, which are rare and hard to design[14–24]. Thus, designing more generalizable methods of creating bistability would be a milestone towards autonomous, evolving systems that can store memory of past events.

Here, we show such a minimal design of memory based on bistability in a chemical reaction cycle without autocatalysis. The chemically fueled reaction cycle activates and deactivates molecules for self-assembly giving rise to chemically fueled crystals. Through a previously observed mechanism[25,26], these crystals exert feedback on their reaction cycle which affects steady state levels. The result is that two possible, stable states can emerge in a continuously fueled reactor, i.e., one with and one without crystals. Even though in such a reactor, all components are turned over on a minute-timescale, each state is indefinitely stable. Moreover, the states can be rapidly switched by transient signals. We combined nine reactors to create a nine-pixel display. Moreover, we connected reactors to use them as switches in an OR-gate. Miniaturization and multiple reaction cycles should give rise to evolvable, complex behavior in future work.

## Results
The design of the bistability is based on a combination of two mechanisms, i.e., chemically fueled crystals that exert feedback on their reaction cycle (Fig. 1a, b) and a phase diagram of those crystals that contains a metastable zone, i.e., a zone where crystals can be present or not, depending on the history of the sample (Fig. 1c)[27]. The chemically fueled crystals are formed by a reaction cycle that has been explored by our group and others[28–33].

N-Boc-protected aspartic acid (precursor 1) is activated by reacting with fuel (1-ethyl-3-(3-dimethylaminopropyl) carbodiimide, EDC) to form its corresponding anhydride (product, Fig. 1a, Supplementary Fig. 1). The anhydride product is transient with a half-life of roughly 35 seconds at 24 °C before it spontaneously hydrolyzes to the precursor (Supplementary Fig. 2a, b), i.e., the deactivation reaction. When fuel and precursor are supplied continuously, a steady state emerges in which the product is activated and deactivated at equal rates (vide infra).

We first determined that phase diagram of the crystallization by using a solution of 100 mM precursor in 200 mM MES buffer at pH 3.5. We added fuel to initiate product formation, and we found that the anhydride product can nucleate to form crystals (crystal structure data is listed in the Supplementary Information) above a supersaturation concentration of 16 mM at 24 °C, i.e., $S_{sat} = 16.0$ mM (Fig. 1c). Vice versa, the crystals dissolve when the concentration product falls below a 7.9 mM ($S_{out} = 7.9$ mM). The concentration range between the $S_{out}$ and $S_{sat}$ is referred to as the metastable zone in the Ostwald-Miers diagram for a solute/solvent system[34]. A solution with a concentration of product in this metastable zone can contain crystals or not depending on whether the concentration of the sample was at some point in the recent history above $S_{sat}$. For example, when 20 mM of EDC is added to 100 mM precursor, a maximum of 12 mM product emerges at the expense of fuel and then decays as fuel is running low (Supplementary Fig. 3). Thus, the system does not reach the $S_{sat}$ and no crystals are observed. Instead, when 60 mM EDC is added, 44 mM product is activated, and crystals are present. When the concentration falls below $S_{out}$ because the system runs low on fuel, the crystals dissolve (Supplementary Fig. 3). A solution that can be sustained within the metastable window, for example, by continuously fueling, could either have crystals or not. However, neither state would be stable indefinitely, as minor fluctuations could result in crystal growth or dissolution.

Thus, a second important feature of the bistability we found is that crystals exert feedback on their reaction cycle. The anhydride in the crystals is protected from hydrolysis because no water can reach the anhydride within the crystal (Fig. 1b). Only the anhydride that remains in the solution can be hydrolyzed, which is equal to the anhydride's solubility ($S_{out}$). Therefore, the presence of crystals severely slows down the overall deactivation rate, resulting in the accumulation of product, and it raises the product concentration outside of the metastable window. The product's self-protection mechanism is effectively a feedback mechanism. Excitingly, the feedback relies on a phase separation mechanism and is, due to its rudimentary nature, frequently observed in

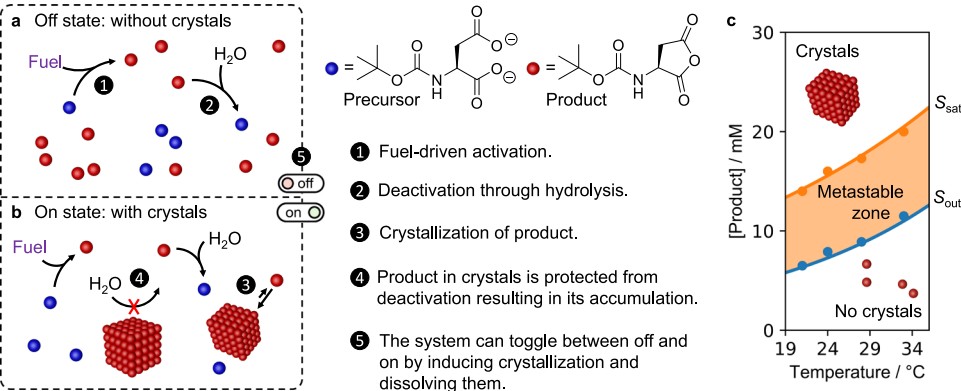

**Fig. 1 Mechanisms for bistability in chemically fueled crystals.** Two possible states can emerge in one experiment either an (**a**), off-state without crystals or (**b**), an on-state with crystals. The crystals protect the anhydride from deactivation which serves as the feedback mechanism. **c** The phase diagram of crystallization of the product. In the metastable window, a solution with or without crystals can exist. Solubility and supersaturation concentrations at various temperatures were experimentally determined (see Supplementary Methods section and Supplementary Fig. 18).

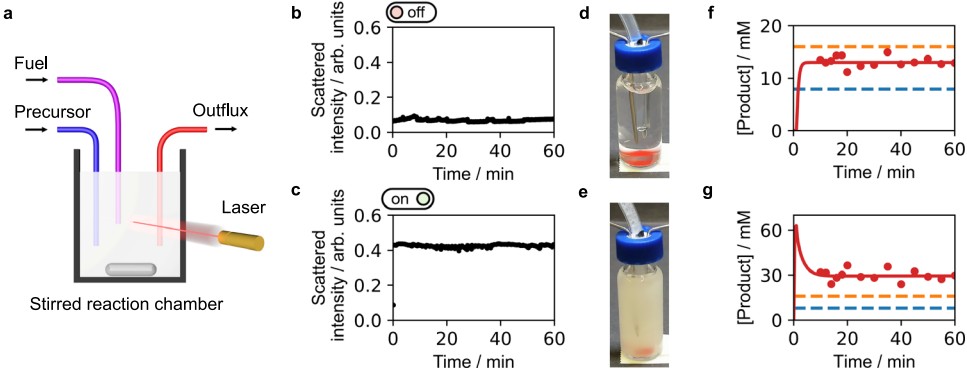

**Fig. 2 Chemically fueled assemblies that exert feedback result in bistability. a** A schematic representation of our experimental setup to keep the product in a steady state concentration. Precursor and fuel are added continuously to a stirred reaction chamber and all the reaction cycle's components are removed by an outflux. **b**, **d** and **f** The off-state was created when a 1.5 ml reactor was supplied with 25 mM.min$^{-1}$ EDC and 30 mM.min$^{-1}$ precursor with a space velocity of 0.4 min$^{-1}$. We confirmed the absence of crystals by (**b**), turbidity and (**d**), a photograph and we measured the steady state concentration by (**f**), HPLC. **c**, **e** and **g** The same experiment as above, but now an 80 mM EDC spike was added in the beginning. The markers in the concentration plot correspond to HPLC data, whereas the red line shows a prediction by our kinetic model. The blue and orange lines are the $S_{out}$ and $S_{sat}$, respectively.

chemically fueled assemblies[35–37]. The metastable zone in the phase diagram combined with the feedback of the crystals are the core mechanism of the bistability in this work. A state without crystals in the metastable zone can be stable. If crystallization is induced, it switches on a feedback mechanism which raises the product concentration outside of the metastable window which is also stable.

We used microsyringe pumps that continuously supplied a 1.5 ml solution in a vial with 0.3 ml.min$^{-1}$ of a 125 mM EDC solution and 0.3 ml.min$^{-1}$ of a 150 mM precursor solution. In other words, the reactor was supplied with 25 mM EDC.min$^{-1}$ and 30 mM precursor.min$^{-1}$. To stop the reactor from over-flowing a pump extracted 0.6 ml.min$^{-1}$, thus creating a steady state (i.e., a continuously stirred tank reactor, Fig. 2a, Supplementary Fig. 4). In the setup, every minute, 40% of the reactor volume was replaced, i.e., the space velocity is 0.4 min$^{-1}$. The sample was placed in a device that constantly monitored its turbidity (Supplementary Fig. 5)[38]. At intervals, the concentration of reactants was measured by HPLC (high-pressure liquid chromatography). The solution remained clear for over an hour (Fig. 2b, d) and HPLC confirmed that the system evolved towards a steady state of 13.0 mM product, i.e., in the metastable zone without crystals (Fig. 2f). If we performed the same experiment but spiked the reactor with 80 mM of EDC just before starting the pumps, the solution became turbid due to the presence of the crystals and evolved to a stable steady state at 29.4 mM, i.e., far above $S_{sat}$ (Fig. 2c, e, g). We determined all rate constants in the chemical reaction cycle (Supplementary Table 1) and combined them in a previously reported kinetic model[26,29,39]. The model takes into account the self-protection mechanism and the metastable window resulting in an accurate prediction of the evolution of the concentration in our experiments (Fig. 2f, g).

The kinetic model was used to predict the steady state anhydride concentration for a range of fuel fluxes to create a so-called hysteresis curve (Fig. 3a). In this curve, the fuel flux is first increased from 0 to 50 mM.min$^{-1}$ and then decreased from 50 to 0 mM.min$^{-1}$ and the corresponding steady state concentrations are calculated. The influx of precursor and the space velocity were kept constant at 30 mM.min$^{-1}$ and 0.4 min$^{-1}$, respectively. From the curve, three regimes emerge. At low energy fluxes, one steady state concentration without crystals is observed independent of whether the system came from a high or low fuel flux. Similarly, at high fuel fluxes, one steady state is found but now with crystals. In between, two possible steady states are found, i.e., a high steady state concentration with crystals, and a low one without.

Experimentally, we validated several points on the hysteresis curve by HPLC (Fig. 3a, Supplementary Fig. 6). To do so, we changed the concentration of the fuel in the solution that feeds the reactor, thus, effectively, changing the influx of fuel with a constant space velocity. After 10 min, the steady state concentration was determined every 2 min. From five datapoints, the mean and deviation in the steady state concentration was calculated. Finally, the turbidity of these samples was also measured to confirm the presence or absence of crystals (Supplementary Fig. 7). We performed similar experiments at various temperatures to qualitatively understand the temperature dependence of the bistable behavior (Fig. 3b, Supplementary Fig. 8–13). We also determined the temperature dependence of the rate constants, $S_{out}$, and $S_{sat}$ (Supplementary Fig. 2, Supplementary Table 1, and Supplementary Fig. 18, respectively), giving us quantitative understanding and allowing us to expand our kinetic model for several temperatures (Fig. 3c). From these curves, it became clear that the bistable window increases with temperature. The reason for the increase was a faster increase of the $S_{sat}$ with temperature compared to $S_{out}$. Effectively, at higher temperatures the metastable zone was larger. Furthermore, deactivation rates are higher at higher temperatures which lowers steady state concentrations. As a result, a higher range of fuel fluxes can be applied with the steady state concentrations within the metastable zone. We found that increasing the space velocity, and thus decreasing the time the reactants remained in the reactor, widened the bistable window as well but required higher energy fluxes for its onset (Fig. 3d, Supplementary Fig. 14–17). In our setup, the loss of product is governed by hydrolysis and by outflow from the reactor. Especially at high space velocities, the outflow of the reactor is dominating. Thus, the effect of the self-protection diminishes, and the steady state concentrations approach each other until no more bistability was observed in the system. Simply put; at high space velocities, the crystals spend too little time in the reactor to store the memory required to sustain the on state. We found that when we decreased the space velocity to 0.0 min$^{-1}$ in our model, i.e., there is only fuel flowing into the reactor, but there is no outflow of precursor, product, or waste, we also observed bistability. These observations are significant because they imply that the bistable signature is inherent to the chemical reaction cycle and not a result of the outflow of product and crystals by pumps. Thus, miniaturization of the system is possible because pumps carrying in precursor and removing the product are not required. For example, the system could be miniaturized by using liposomes that are constantly supplied with fuel through

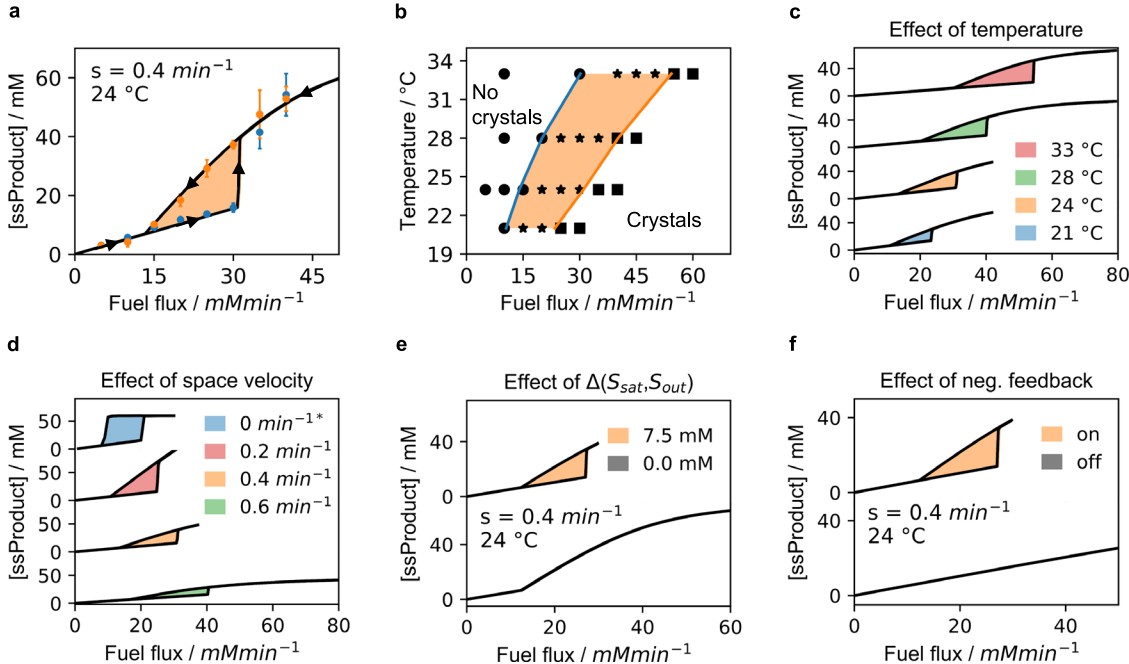

**Fig. 3 The bistable window can be tuned by a range of parameters. a** The steady state concentration determined by the kinetic model (line) and HPLC (markers) at different fuel fluxes yields a hysteresis curve. The precursor influx was kept at 30 mM.min$^{-1}$ and the space velocity was 0.4 min$^{-1}$. The markers represent the measured mean steady state concentration and their standard deviation ($n = 5$). **b** The bistability window at various temperatures as determined by turbidity measurements. The conditions are the same as for a except for the temperature. The stars mark the energy flux at which two stable steady states were found. **c** The effect of temperature on the hysteresis curve as determined by the kinetic model. The conditions are the same as for b. The intersection of the high- and low energy branch is drawn in b as blue and orange line. **d** The effect of the space velocity on the bistability window as calculated by the kinetic model at 24 °C. The conditions are the same as for a. (*) For a space velocity of 0.0 min$^{-1}$, 60 mM precursor starting concentration was used. **e** The influence of the metastable zone and (**f**), negative feedback on the bistability behavior.

nanopores[40]. Finally, the kinetic model confirmed the two physicochemical mechanisms that were critical for the bistability to function, i.e., the metastable window for crystallization, and the crystal's self-protection mechanism. We calculated the hysteresis curve when $S_{out}$ was equal to the $S_{sat}$ (metastable window = 0.0 mM) and compared it to a hysteresis curve with the measured metastable window of 7.5 mM (Fig. 3e). In the curve without a metastable window, the steady state concentration rises with increased EDC flux until the solubility limit is reached. Then, the further increase in EDC flux results in a sudden increase in steady state concentration because crystals are formed that protect the anhydride from hydrolysis, but no hysteresis is found. The importance of the metastable window could be verified experimentally using chemically fueled droplets, which also protect their anhydride through phase separation[26]. However, unlike the crystals, they do not have a measurable metastable window, i.e., their $S_{sat}$ equals $S_{out}$ at 3.5 mM for 2-hexen-1-yl-succinic acid (precursor **2**)[41]. Our kinetic model verified that no hysteresis was found for the droplet-based system, which was experimentally confirmed (Supplementary Figs. 19 and 21). Next, we tested the importance of the self-protection mechanism for the bistability, i.e., we switched off the mechanism in the kinetic model such that the deactivation could occur on both the anhydride in solution and on the anhydride in the crystals with equal rate constants. Without feedback, no bistable zone was found (Fig. 3f).

With the fundamental understanding of the bistability, we tested the response of another precursor (N-Boc-protected glutamic acid, precursor **3**) to various fuel fluxes. This precursor is similar in structure but differs in its reaction rates (Supplementary Fig. 2d and Supplementary Table 1). Given its high solubility, 2 M of NaCl was added to reduce the amount of fuel needed to overcome the supersaturation concentration and initiate crystal formation. A similar hysteresis curve was found compared to precursor **1** (Supplementary Figs. 20 and 21).

Next, we tested the bistable system to serve as volatile memory, i.e., its ability to convert a transient input to a sustained response. First, we sought transient inputs. An off-state was created at 21 °C by feeding a solution 20 mM EDC.min$^{-1}$ and 30 mM precursor.min$^{-1}$ at a space velocity of 0.4 min$^{-1}$. We used a batch of EDC as input and, with 20 mM EDC or more, the surge in EDC induced crystal formation and switched the system to the on state (Supplementary Fig. 22a). We monitored by turbidity the time it took to switch on the system (Fig. 4a). Unsurprisingly, the more fuel is added the faster the system is switched on. Noteworthy, the steady state level in EDC restored within 2 min while the anhydride steady state level sustained (Supplementary Fig. 22b). We tested several other transient inputs and found the addition of product itself was the most efficient (Fig. 4a, Supplementary Fig. 22c). As little as 0.1 mM of product **1** was required to switch on the system with a half-life time of roughly seven minutes. The addition of high concentration of salts could also serve as input, likely because it transiently decreased $S_{Sat}$ of the system, i.e., it effectively salted out the product (Supplementary Fig. 22d–f). That was further confirmed by the fact that the half-life time followed Hofmeister's series of salting out proteins[42]. Similarly, we sought inputs that could transiently perturb the steady state such that it remained in an off state. We tested several primary amines as input because they react with the product to form their more soluble amide (Fig. 4b and Supplementary Fig. 23). All amines we tested were able to switch off the bistable state, albeit with varying effectiveness. Benzylamine, being the most of them all, allowed us to turn the system off within few seconds. To demonstrate the reversibility of the system, we toggled between on and off several times using 40 mM EDC and 300 mM

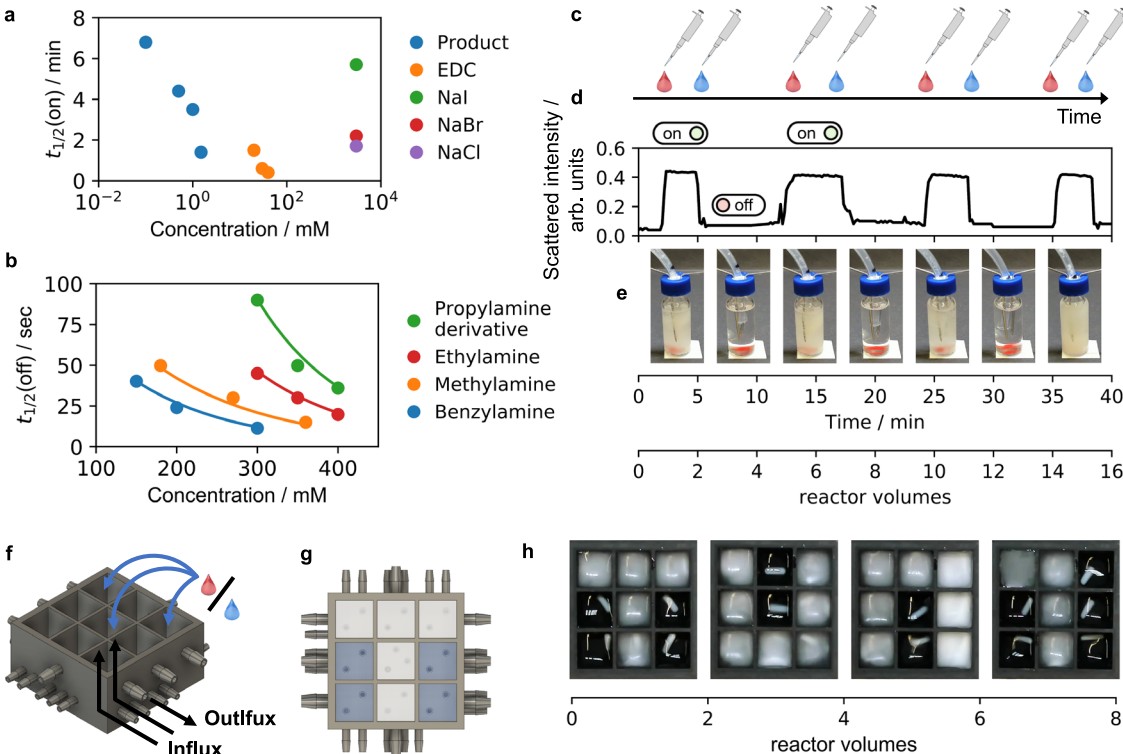

**Fig. 4 Toggling between on and off can create pixel displays.** The half-life of switching from (**a**), off to on and (**b**), on to off when different triggers are added to the reactor vial. The experimental conditions are 30 mM precursor.min$^{-1}$ and 20 mM EDC.min$^{-1}$ flux at 21 °C and a space velocity of 0.4 min$^{-1}$. **c** The system is switched between states several times using EDC (red drop) and benzylamine (blue drop) as triggers. **d** The system's turbidity over time as measured by scattered light. **e** Time-lapse photographs of the bistable system. **f** A 3D printed 3 × 3 array in which each reactor can be switched individually and (**g**), the top view picturing a pixel display. **h** By switching particular reactors, TUM is printed.

benzylamine as inputs (Fig. 4c–e and Supplementary Movie 1). Excited about the system's visible ability to store memory, we extended the concept to a matrix display. We developed a matrix of 3 by 3 continuously stirred tank reactors (Fig. 4f, g and Supplementary Fig. 24). Each was continuously fueled with the same conditions as described above. Switching on the pumps gave clear solutions in all pixels. Next, individual reactors were spiked with fuel to print characters (Fig. 4h). Indeed, crystals appeared and sustained which made the pixel switched from off (black) to on (white) resulting in the letter T. Next, we converted the letter T into the U by switching off pixels by addition of benzylamine and switching on others by spiking with fuel. Each reactor could be addressed individually allowing us to print and erase 9-pixel-based patterns or digits (Supplementary Movie 2).

Finally, we explored the idea of connecting switches to create a logic gate[43,44]. In such devices, the output of the gate is dependent on the state of multiple inputs. We opted here for a simple logic gate, i.e., the OR-port (Fig. 5a, b). It must be noted that the OR-port is the most straightforward gate using our design. The design of communicating reactors can, in principle, be further expanded to an AND-port but not to other logic gates in which a positive input should lead to a negative output (like a NOT or NOR-port). In our OR-port, the state of a readout reactor which we call output, is dependent on the state of multiple input reactors. The relation between input and output can be read in a truth table (Fig. 5c). For example, in an OR-port, the output is positive if one input or both inputs are positive. The output is negative only when both inputs are negative. We 3D-printed a design that contains three reactors that can individually be supplied with fuel and precursor (Fig. 5d). The outer two reactors are the input reactors and act as switches, i.e., various triggers (e.g., salt, EDC, or amines) can switch them on or off. The middle reactor is our

readout reactor and can compute based on the state of the input reactors. We connected the input reactors to the output reactor through a small hole in the walls (Fig. 5e). Moreover, each of the input reactors had a lower outflow (0.4 ml.min$^{-1}$) compared to inflow (0.6 ml.min$^{-1}$). Vice versa, the middle output reactor had a lower inflow (0.2 ml.min$^{-1}$) compared to outflow (0.6 ml.min$^{-1}$). Consequently, 0.2 ml.min$^{-1}$ of the liquid from each input reactor was forced through the output reactor (Supplementary Fig. 25). Taken together, the input reactors were regular bistable switches and served as input for the output reactor. Using time-lapse photography and image analysis software, we monitored the outflow of each reactor to quantify the response of different input combinations. When we switched on the system, input A and input B were both off, and the middle output reactor remained clear, too (Fig. 5f, Supplementary Movie 3). Upon switching input A on by adding 40 mM EDC, the output switched from off to on, which was confirmed by a rise in the calculated mean grey value. Input B was not affected by this conversion and stayed off. After 10 min, reactor A was switched off by the addition of 300 mM benzylamine. The output also switched off, albeit somewhat slower (Supplementary Fig. 25). Next, we activated input B by the addition of 3 M NaCl. Again, the change in the input signal B, switched on the output. By assigning a digital 1 to states above a mean grey value threshold of 30 arbitrary units and a digital 0 to states below, we met the conditions required for a logic OR-port[45]. These final experiments demonstrate that it is possible to create an OR-port using our crystal-based bistable reaction network.

## Discussion
We showed that bistability can emerge from a fuel-driven reaction cycle that forms crystals. The simplicity means we can now

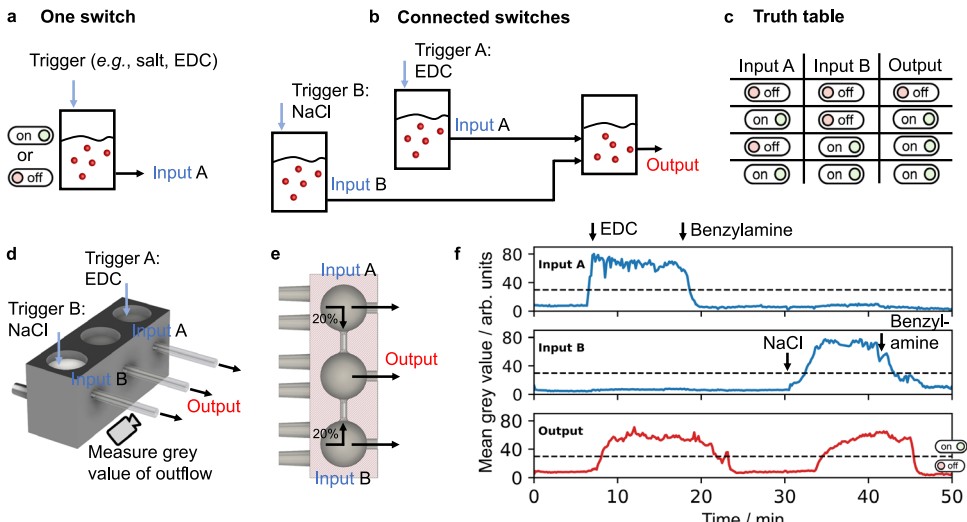

**Fig. 5 An OR-gate through the combination of bistable switches. a** A schematic representation of a switch. The outflow of the reactor is envisioned as the input for a logic gate. **b** A schematic representation of a logic gate. Two switches affect the state of a readout reactor. **c** The truth table of an OR-port. **d** The 3D printed setup of three connected reaction chambers. **e** Top view section analysis of the printed setup. Part of the input reactors flow into the output reactor. **f** The measured grey value over time detected from the outflux tubing of the three reaction chambers. The experimental conditions are 30 mM precursor.min$^{-1}$ and 20 mM EDC.min$^{-1}$ at 21 °C and a space velocity of 0.4 min$^{-1}$.

produce memory in systems chemistry without the need for autocatalytic networks or enzymes. The bistable state can be toggled from on and off by small-molecule inputs allowing us to engineer a pixel-based device which can be printed and erased. Like in biological signaling cascades, we coupled reactors such that multiple inputs are used to compute an output. We seek to miniaturize the behavior, for example, in microfluidic droplets or vesicles. We envision coupling each state to a physical response of the compartment by making use of the crystallization. For example, the crystallization could induce deformation or even self-division of the compartment, such that, when several conditions are met, the compartment can make the decision, through Boolean logic, to self-divide.

## Methods
**Materials**. 1-ethyl-3-(3-dimethylaminopropyl)carbodiimide (EDC), NaCl, NaBr, NaI, benzylamine, 33 wt% methylamine in absolute ethanol, 66.0–72.0% ethylamine in H$_2$O, 3-(Dimethylamino)-1-propylamine, latex beads (polystyrene, 0.1 μm mean particle size), 4-morpholineethanesulfonic acid (MES) buffer were all purchased from Sigma–Aldrich. The precursor N-Boc-L-aspartic acid **1** and N-Boc-L-glutamic acid **3** were purchased from Alfa Aesar and TCI, respectively. All chemicals were used without any further purification. The precursor 2-hexen-1-yl-succinic acid **2** was synthesized by treating the corresponding anhydride with two (mass)-equivalents of MQ water, stirred overnight at room temperature, and then lyophilized[41]. The product was stored at −20 °C until further use. The anhydride of N-Boc-L-aspartic acid was synthesized according to a previously published procedure, treating the diacid with the carbodiimide coupling agent DCC and recrystallization from an acetone/hexane mixture[30]. HPLC grade acetonitrile was purchased from VWR. Milli-Q water was used provided from a Millipore water purifier system.

**General sample preparation**. We prepared stock solutions of the precursors by dissolving them in 0.2 M MES buffer, after which we adjusted the pH to 3.5 and 6.0 for precursor **1**, **3** (additional 2 M NaCl) and **2**, respectively. Stock solutions of EDC and benzylamine were prepared freshly by dissolution in MQ water.

**Batch-fueled experiments**. A 100 mM stock solution of precursor **1** or **3** was put in a 1.5 ml HPLC vial equipped with a micro-stir bar. To this solution, various amounts of EDC were added from a 2 M stock solution. The dilution of the precursor solution was kept below 5%. The temperature was adjusted by the Mini Inkubator from Labnet International. The reaction cycle's concentrations were monitored over time by HPLC as described below. These experiments were performed at various temperatures. From these concentrations, we determined the rate constants of deactivation. The remaining rate constants of the reaction cycle were fitted with the kinetic model (vide infra).

**HPLC to determine the concentration in the reaction cycle**. We applied analytical HPLC (HPLC, Dionex/Thermo Fisher Scientific Ultimate 3000, Hypersil Gold, reversed phase C$_{18}$ column, particle size: 5 μm, length: 250 mm, ID: 20 mm). At several timepoints 30 μl aliquots of the reaction mixture were taken and quenched by 30 μL of a 200 mM benzylamine in water solution[46]. Samples (2 μl injections) were eluted by a linear gradient of acetonitrile in water (Supplementary Fig. 26). Calculation of concentrations was done applying standard curves[46].

**Turbidity measurements**. The evolutions of the turbidity during the reaction cycle are measured with a home-made setup[38]. The Arduino controlled setup measured transmittance by the detection of scattered laser light. A detailed description of the setup is given in the Supporting Information.

**Kinetic model**. Calculation of the reaction cycle's concentrations in batch fueled and continuously fueled experiments was done by a kinetic model described in previous work[29,39,46]. Briefly, a set of five differential equations determines the reaction rates of the direct hydration of EDC to yield EDU ($r_0$), the reaction of precursor with EDC to yield O-acylisourea and EDU ($r_1$), the hydrolysis of the O-acylisourea to yield precursor and EDU ($r_3$), the conversion of the O-acylisourea to yield anhydride and EDU ($r_2$), and the hydrolysis of anhydride to yield the precursor ($r_4$). From these rates, the concentration of EDC, precursor, O-acylisourea, EDU, and anhydride is calculated every second. All reactions are drawn out in Supplementary Fig. 1. To implement the self-protection mechanism, we added to the kinetic model a statement that determines whether crystals are present or not. Crystals are present if the concentration exceeded the supersaturation concentration ($S_{sat}$). The crystals dissolve when the concentration falls below the solubility ($S_{out}$). When crystals are absent, the rate of hydrolysis is equal to the rate constant * [anhydride] (1st order). When crystals are present, the rate of hydrolysis if equal to the rate constant * $S_{out}$ (0th order), because the crystals protect all anhydride except for the anhydride that remains in solution ($S_{out}$). Thus, the model switches from first order to zero order deactivation rates and vice versa, when crystals start to form and dissolve, respectively. The models are provided at: https://github.com/fabianschnitter/Bistability-in-a-chemical-reaction-network.git.

**Steady state experiments**. We used a continuously stirred tank reactor to create steady state concentrations. In brief, two syringe pumps continuously supplied an HPLC vial with fuel and precursor. A third pump withdrew liquid from the reactor to prevent overflowing. The contents of the vial were stirred by a micro-stirrer. A detailed explanation of the experimental method is given in the Supporting Information.

**Determining the temperature dependence of $S_{out}$ and $S_{sat}$**. The phase diagram was drawn by determining $S_{out}$ and $S_{sat}$ in steady state experiments. Regarding $S_{out}$ the system was spiked, and an EDC flux was chosen unable to sustain the on state. The anhydride concentration at the timepoint of crystal dissolution determined by scattered light was chosen as $S_{out}$. The supersaturation concentration $S_{sat}$ was estimated by steadily increasing the EDC flux until sustained crystal formation was

detected. A detailed explanation of the setup is given in the Supporting Information.

**Toggling the switch experiments**. The switch was turned to on from a steady state concentration inside the metastable zone by addition of several triggers. The reactor was switched off by addition of primary amines. A detailed description of the experimental method is given in the Supporting Information.

**Pixel display experiment**. The $3 \times 3$ pixel display reactor was 3D printed. Each of the nine reactors was filled with 0.6 ml of 100 mM precursor **1**. Then, the flowrate in was set to 0.12 ml.min$^{-1}$ (150 mM acid stock, 100 mM EDC stock) and the flowrate out to 0.24 ml.min$^{-1}$. Activation of the pixels was done by the addition of 12 µl of a 2 M EDC stock, whereas deactivation was done by replacing 36 µl of the reaction solution with 36 µl 5 M benzylamine solution. A detailed description of the experimental method is given in the Supporting Information.

**OR-port experiment**. A $1 \times 3$ array was 3D printed. The flow from the input to the output reactor takes place via a 2 mm diameter hole. The reactor's turbidity is monitored by time lapsed photographs of the outflow tubings in 10 s intervals. Each reactor was filled with 1.5 ml water. Then, each reactor received an influx of 20 mM.min$^{-1}$ EDC and 30 mM.min$^{-1}$ precursor. A detailed description of the experimental method is given in the Supporting Information.

## Data availability

The authors declare that all the data supporting the findings of this study are available within the article, the source data files and the Supplementary Information files. Source data are provided with this paper.

## Code availability

The Matlab code used to calculate the reaction cycles' concentrations and the script controlling the turbidity device are provided through repositories at github and zenodo. The respective links and references are provided with this paper.

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

## Acknowledgements

The BoekhovenLab is grateful for support by the TUM Innovation Network - RISE funded through the Excellence Strategy. This research was conducted within the Max Planck School Matter to Life supported by the German Federal Ministry of Education and Research (BMBF) in collaboration with the Max Planck Society. F.S. and B.R. acknowledge funding by the Deutsche Forschungsgemeinschaft (DFG, German Research Foundation) – SFB-863 – Project ID 111166240 (Project B11).

## Author contributions

F.S. designed and performed the experiments and wrote the manuscript. B.R. built and wrote the code for the turbidity device. C.J. performed the single crystal X-ray diffraction experiment and evaluated the crystal structure data. J.B. designed experiments, outlined, and wrote the manuscript, and supervised the project. All authors discussed the experimental results.

## Funding

## Competing interests

The authors declare no competing interests.
