## [Peer Review File · Nature Communications]

Memory, switches, and an OR-port through bistability in chemically fueled crystalsEditorial Note: This manuscript has been previously reviewed at another journal that is not operating a transparent peer review scheme. This document only contains reviewer comments and rebuttal letters for versions considered at *Nature Communications*.

REVIEWER COMMENTS

Reviewer #1 (Remarks to the Author):

I have now gone through all the point by point response of my comments and the other reviewer comments and the corrections made. I support the publication of this revised manuscript.

Reviewer #2 (Remarks to the Author):

The new version of the paper by Boekhoven and coworkers contains some new discussion, and clarification for certain topics.

- The kinetic model is better explained and referenced now.
- My requests to add new data – see comments #5 (HPLC data) and #6 (an example of bistability with a compound more differing from Boc-glu) – were only addressed by discussion items; This reflects a compromised scholar presentation, but I would leave it to the authors to decide what else can be provided in the next cycle.
- As for my original comment #7, in light of a similar comment by reviewer #1, I still hold the opinion that the section on logic gates should be dramatically expanded, or otherwise removed all together or at least significantly downplayed (by for example taking it out from the title, abstract etc.).

COMMENTS TO THE AUTHOR. Round 2.

Reviewer #1 (Remarks to the Author):

I have now gone through all the point by point response of my comments and the other reviewer comments and the corrections made. I support the publication of this revised manuscript.
Thank you for the time you have invested in improving our manuscript.

Reviewer #2 (Remarks to the Author):

The new version of the paper by Boekhoven and coworkers contains some new discussion, and clarification for certain topics.

Thank you for the time you have invested in improving our manuscript.

- The kinetic model is better explained and referenced now.
- My requests to add new data – see comments #5 (HPLC data) and #6 (an example of bistability with a compound more differing from Boc-glu) – were only addressed by discussion items; This reflects a compromised scholar presentation, but I would leave it to the authors to decide what else can be provided in the next cycle.

Original comment 5. The flow reactor data for 24C includes the system's characterization by direct analysis of the product amounts (HPLC; fig. 3a & s6) and the formation of crystals (scattering; fig s7). For all other temperatures, only the semi-quantitative analysis by scattering was applied. Is there a special reason for that? Can the HPLC analysis for the other samples be completed to support the data in Fig 3b?

We apologize. We misunderstood the question. We now added further HPLC data for each of the four temperatures we tested (Supplementary Fig. 8, 10 and 12). They support the temperature-dependence of the bistability window and further corroborates our initial findings based on the scattering data. In summary, at all four temperatures, we see a bistable window. We see the bistable window is temperature dependent. Finally, we validated that our kinetic model is also accurate at the other three temperatures.

Original comment 6. In page 9, the paper describes the data with Boc-Glu, for which similar behavior was observed. The authors then claim that this shows the 'generality' of the system.... A better choice to demonstrate the generality, would of course include a compound that differs from Boc-Asp much more (different sequence, different reaction rate, different concentration regime for phase separation etc.). This claim should be downplayed.

Regarding the request on Boc-Glu, we downplayed the argument as you requested in the first request for revisions. We now mention its structural similarity compared to precursor **1** but point at its differing reaction rates (slower ring closure k_2 and slower deactivation kinetics k_4).

- As for my original comment #7, in light of a similar comment by reviewer #1, I still hold the opinion that the section on logic gates should be dramatically expanded, or otherwise removed all together or at least significantly downplayed (by for example taking it out from the title, abstract etc.).

We chose to further downplay the section. We have removed any mentioning of logic gates (also in the title). We added an explicit mention that this concept of the OR-port cannot be further explored beyond an AND-port.

REVIEWER COMMENTS

Reviewer #2 (Remarks to the Author):

I am satisfied by the changes made to the manuscript.
I can support publication in the current form.

COMMENTS TO THE AUTHOR. Round 3.

Reviewer #2 (Remarks to the Author):

I am satisfied by the changes made to the manuscript.

I can support publication in the current form.

Thank you for the time you have invested in improving our manuscript.